# Protaetia Brevitarsis-Derived Protein Hydrolysate Reduces Obesity-Related Colitis Induced by High-Fat Diet in Mice through Anti-Inflammatory Pathways

**DOI:** 10.3390/ijms241512333

**Published:** 2023-08-02

**Authors:** Hyung Jun Kwon, So Young Chun, Eun Hye Lee, BoHyun Yoon, Man-Hoon Han, Jae-Wook Chung, Yun-Sok Ha, Jun Nyung Lee, Hyun Tae Kim, Dae Hwan Kim, Tae Gyun Kwon, Bum Soo Kim, Syng-Ook Lee, Byung Ik Jang

**Affiliations:** 1Department of Surgery, School of Medicine, Kyungpook National University, Daegu 41566, Republic of Korea; kwonhj95@knu.ac.kr; 2BioMedical Research Institute, Kyungpook National University Hospital, Daegu 41944, Republic of Korea; 3Joint Institute for Regenerative Medicine, Kyungpook National University, Daegu 41566, Republic of Korea; 4Department of Pathology, School of Medicine, Kyungpook National University, Daegu 41566, Republic of Korea; 5Department of Urology, School of Medicine, Kyungpook National University, Daegu 41566, Republic of Korea; jeus119@hanmail.net (J.-W.C.); ljnlover@gmail.com (J.N.L.); tgkwon@knu.ac.kr (T.G.K.); urokbs@knu.ac.kr (B.S.K.); 6Department of Laboratory Animal Research Support Team, Yeungnam University Medical Center, Daegu 42415, Republic of Korea; ikorando5@hanmail.net; 7Department of Food Science and Technology, Keimyung University, Daegu 42601, Republic of Korea; 8Department of Internal Medicine, School of Medicine, Yeungnam University, Daegu 42415, Republic of Korea

**Keywords:** colitis, P. brevitarsis, anti-inflammatory response, protein hydrolysate, high-fat diet

## Abstract

Ulcerative colitis is an inflammatory bowel disease characterized by inflammation in the mucosal and submucosal layers of the colon. Obesity is closely related to the occurrence and progression of colitis. The most plausible mechanism linking obesity and colitis is an excessive adipogenesis-related inflammatory response, which causes mucosal dysfunction. Obesity and colitis are linked by several etiologic mechanisms, including excessive adipogenesis, lipotoxicity, pro-inflammatory adipokines/cytokines, macrophage polarization, oxidative stress, endoplasmic reticulum (ER) stress, and gut microbiota. These low-grade enteric inflammations cause mucosal layer damage, especially goblet cell dysfunction through mucin 2 (MUC2) misfolding, ultimately leading to colitis. Inhibiting the inflammatory response can be the most effective approach for treating obesity-related colitis. We focused on the anti-inflammatory effects of polyphenols in Protaectia brevitas larvae. The P. brevitas was prepared as a low molecular protein hydrolysate (PHPB) to increase the concentration of anti-inflammatory molecules. In the current study, we investigated the anti-inflammatory effect of PHPB in an obesity-induced colitis mouse model. Compared with the high-fat diet (HFD) group, the group treated with PHPB exhibited reduced body/organ/fat weight, appetite/food intake inhibition, hypolipidemic effect on ectopic fat, and anti-adipogenic mechanism through the AMPK signaling pathway. Furthermore, we observed attenuated expression of PPARγ and C/EBPα, inhibition of pro-inflammatory molecules, stimulation of anti-inflammatory molecules, probiotic-like effect against obesogenic gut microbiota, inhibition of macrophage polarization into M1, suppression of oxidative/ER stress, and reduction of Muc2 protein misfolding in colon. These diverse anti-inflammatory responses caused histological and functional recovery of goblet cells, eventually improving colitis. Therefore, our findings suggest that the protein hydrolysate of Protaetia brevitarsis can improve obesity-related colitis through its anti-inflammatory activities.

## 1. Introduction

Ulcerative colitis is an inflammatory bowel disease that causes inflammation in the mucosal and submucosal layer of the colon [1]. Obesity is closely related to the occurrence and progression of colitis [2], with the excessive adipogenesis-related inflammatory response being the most plausible mechanism causing mucosal dysfunction [3].

Obesity-colitis-linked etiologic mechanisms are related to the following pathways: (1) Excessive adipogenesis through the AMPK phosphorylation-PPARγ-C/EBPα pathway [4]; (2) Lipotoxicity by free fatty acid through the ERK-JNK-p38-IκBα-NF-κB pathway [5]; (3) Lipopolysaccharide toxicity through the TLR4-MyD88-JNK/NF-κB pathway [6]; (4) enhanced secretion of pro-inflammatory adipokines/cytokines by adipocytes/immunocytes [7,8]; (5) inhibited secretion of anti-inflammatory adipokines/cytokines [7,8]; (6) macrophage polarization into M1 rather than M2 [9]; (7) increased oxidative stress through the activation of iNOS and COX-2 enzymes, resulting into nitric oxide and prostaglandin E2 synthesis [3]; (8) endoplasmic reticulum (ER) stress with increased expression of sXbp1, Grp78, and Edem1 proteins [3]; (9) exosome [10]; (10) micro RNA10; and/or (11) gut microbiota [11]. Low-grade enteric inflammation damages the mucosal layer, especially goblet cell dysfunction, through mucin 2 (MUC2) protein misfolding [3]. This is the primary mechanism associated with colitis development. MUC2 is an essential protein associated with oligomeric mucus/gel-forming [12]; thus, MUC2 misfolding causes abnormal mucin biosynthesis and increases mucosal layer permeability. The weakened and thinned mucosal barrier cannot effectively protect the epithelial layer and the area beneath the lamina propria against triggers, which results in colitis [3].

Based on the etiology of obesity-related colitis, the inhibition of inflammatory response can be the most effective treatment approach. Previous studies have focused on the anti-inflammatory effects of natural molecules, especially polyphenols [13]. Phenolic compounds, represented by flavonoids, have antioxidant, free-radical scavenger, metal-ion chelating, vasoprotective, hepatoprotective, anti-inflammatory, anti-cancer, anti-infective, and anti-diabetic properties [14]. Protaectia brevitas larvae contain substantial amounts of polyphenols [15]. Polyphenol and flavonoid contents of P. brevitas range between 19.2–30.4 mg/g and 0.16–0.23 mg/g, respectively [16]. We prepared P. brevitas as a low molecular protein hydrolysate (PHPB) using an optimized enzyme (alcalase) to increase the concentration of the anti-inflammatory molecules [15]. In vitro studies demonstrated the effective radical scavenging, antioxidant, and anti-inflammatory properties of the prepared PHPB [15].

In the current study, we investigated the anti-inflammatory effect of PHPB in an obesity-induced colitis mouse model through colon length, colon/adipose tissue weight, adipogenesis, pro-/anti-inflammatory cytokine/adipokine, macrophage polarization, oxidative/ER stress, gut microbiota, and goblet cells analyses. The results of these analyses will provide an understanding of the interaction among obesity, colitis, and PHPB. Based on the disease pathogenesis, phenotypic expression, and response to therapy, we expect that PHPB can be an effective treatment for obesity-related colitis.

## 2. Results

### 2.1. Effects of PHPB on Stools, Colon Length/Weight, and Immune-Related Organs

To verify HFD-induced colitis, colon states were observed based on the modified disease activity index [17] (Figure 1A). The Control group exhibited normal stool consistency and negative gross rectal bleeding. The HFD group presented eight cases of loose stools and three cases of hemoccult. The PHPB group exhibited four cases of loose stools; however, hemoccult was not observed. Although frequent soft stools and hemoccult were observed with HFD, PHPB treatment improved stool consistency and prevented rectal bleeding.

The mean colon lengths in the Control, HFD, and PHPB groups were 7.45 ± 0.25 cm, 6.45 ± 0.21 cm, and 6.72 ± 0.34 cm (Figure 1A,B), respectively. As the shortened colon length can be a predictor for colitis, the pathologist confirmed HFD-induced colitis in the HFD group. The PHPB group exhibited a relatively less shortened colon length compared with that of the HFD group.

The HFD-induced colitis was also evidenced by the colon weight (Figure 1C). The proportion of length and weight of the colon is related to inflammatory response or edema. The HFD group exhibited increased weight (0.16 ± 0.02 g) than the PHPB group (0.11 ± 0.01 g). As with colon weight, other immune-related organs also showed weight change patterns (Figure 1D). With HFD, the weights of the spleen, lung, kidney, and liver increased (0.08 ± 0.01, 0.27 ± 0.04, 0.46 ± 0.04, and 1.92 ± 0.08 g, respectively) compared with that of the normal diet (0.07 ± 0.01, 0.18 ± 0.02, 0.37 ± 0.03, and 1.31 ± 0.05 g, respectively). With PHPB treatment, the weights of spleen, lung, and kidney were slightly decreased (0.06 ± 0.01, 0.20 ± 0.02, and 0.43 ± 0.04, respectively) compared to the HFD group, and liver weight was significantly decreased (1.3 ± 0.09 g) (*p* < 0.05).

### 2.2. Effects of PHPB on Body Weight and Food Intake

The body weights of the mice in the Ctrl, HFD, and PHPB groups on the last day of the experiment were 31.97 ± 3.43, 45.89 ± 3.51, and 35.02 ± 3.77 g, respectively (*p* < 0.05) (Figure 2A). The rate of weight gain in the Ctrl group was moderate (about 1 g per week), whereas the HFD group demonstrated a two-fold increase in weight. The increase in the body weight in the PHPB group exhibited the same pattern as that in the HFD group in the first three weeks; however, the weight increase rate decreased to 0.5 g/week after week 4. The net increase in weight in the Ctrl, HFD, and PHPB groups was 10.37 ± 2.87 g, 24.99 ± 2.91 g, and 13.92 ± 3.34 g, respectively (Figure 2B). The gained net weight in the PHPB group was significantly reduced compared with that in the HFD group (*p* < 0.05).

Furthermore, comparing the volumes of food intake in the different groups suggested their direct relation to the body weights (Figure 2C). The mean intake in the Ctrl group was 187.50 ± 20.79 g/week. The HFD and PHPB groups exhibited a significantly reduced mean food intake (144.00 ± 15.64 and 126.60 ± 20.46 g/week, respectively) compared with the Ctrl group (*p* < 0.05), and the PHPB group showed more reduced value than the HFD group.

The volume of food intake was converted into calories (Figure 2D) owing to the differences in their calorie contents (2.93 kcal/g for standard diet and 5.24 kcal/g for HFD). The mean calorie of the Ctrl group was 549.38 ± 60.90 kcal for 10 weeks. Concordantly, the HFD group exhibited a significantly increased calorie content (754.56 ± 81.94 kcal) compared with that in the control (*p* < 0.05). The PHPB group presented more reduced mean calorie content (663.38 ± 107.22 kcal) compared to the HFD group.

### 2.3. Effects of PHPB on Fat Accumulation, Lipid Metabolism, and Adipogenesis

The excessive calories were commonly stored in the abdomen as fat tissues (Figure 3A). The HFD group demonstrated excessively developed adipose tissue in the abdominal cavity; the liver had an evident inflammatory scar on the surface, the lungs and kidneys were pale, and heart and spleen were thickened. However, the PHPB group exhibited a low fat mass in the cavity, and the target organs had a close-to-normal appearance. To define the anti-adipogenic effect of PHPB, the weights of the abdominal fats, such as visceral, epididymal, and perirenal fats, were measured (Figure 3B). The PHPB group demonstrated a low fat weight compared with the HFD group, and the value was significant for the epididymal adipose tissue (*p* < 0.05).

To explain the low volume of abdominal fat by PHPB, adipogenesis-related pathways, serum levels of lipid metabolites, and the expression of AMPK-related genes were analyzed. The PHPB-treated group demonstrated relatively decreased T-chol, TG, and LDL levels (166.4 ± 11.38, 505.8 ± 141.86, and 12.26 ± 112 mg/dL, respectively) and an increased HDL level (87.74 ± 6.29 mg/dL) (Figure 3C).

To explain inhibited systemic lipid metabolites, the adipogenesis-related AMPK pathway was analyzed with visceral adipose tissue. Excessive adipogenesis is correlated to the AMPK pathway, activating PPARγ and C/EBPα protein synthesis in obese conditions. In Western blot analysis, the normalized intensity values for PPARγ were Ctrl, 42,206.65; HFD, 62,705.40; PHPB, 54,545.87, for C/EBPα, Ctrl, 41,486.67; HFD, 49,553.02; PHPB, 27,153.72, for p-AMPKα, Ctrl, 12,847.24; HFD, 33,228.99; PHPB, 52,847.24, for AMPKα, Ctrl, 43,653.19; HFD, 54,615.45; and PHPB, 49,170.63 (Figure 3D). PPARγ and C/EBPα transcription factors synthesis was reduced with PHPB treatment, whereas p-AMPKα protein concertation was significantly increased compared with that in the HFD group (*p* < 0.01).

### 2.4. Effects of PHPB on Inflammatory Molecules

Excessive lipid metabolism is closely associated with increased inflammatory responses. Thus, we analyzed the effect of PHPB on the secretion of inflammatory cytokines/adipokines. To analyze the systemic anti-inflammatory effect of PHPB against low-grade inflammatory conditions induced by HFD, the serum levels of cytokines/adipokines were measured. Compared with the HFD group, the PHPB group exhibited relatively low serum levels of pro-inflammatory IL-6, TNF-α, IGF-1, and leptin (19.63 ± 0.50 ng/mL, 263.87 ± 15.28 pg/mL, 25.69 ± 1.39 pg/mL, and 26.666 ± 5120.19 ng/mL, respectively). In contrast, the concentration of anti-inflammatory adiponectin was high (85.25 ± 0.49 ng/mL) (Figure 4A).

Next, the local inflammatory effect was analyzed with visceral adipose and colon tissues. In visceral adipose tissue, the expressions of pro-inflammatory TNF-a, IL-1β, IL-6, and IL-17a genes in the PHPB group were relatively low compared to the HFD group. Conversely, the expressions of anti-inflammatory adiponectin, IL-10, defensin, and lipin genes were relatively high (Figure 4B). The anti-inflammatory effect of PHPB was also observed in the colon tissue (Figure 4C).

### 2.5. Effects of PHPB on Gut Microbiota and Serum LPS

To investigate the effect of PHPB on gut microbiota in relation to inflammatory response, the DNA expression of vital species of the gut flora and serum LPS level was measured. The DNA expression of Gram-negative Bacteroides-Prevotella spp. and Gram-positive Lactobacillus spp. in the cecum was evaluated (Figure 5A). The decreased DNA synthesis of Bacteroides-Prevotella spp. by HFD (13.47 ± 0.63) was significantly increased by PHPB treatment (16.87 ± 1.09) (Ctrl, 15.45 ± 1.67) (*p* < 0.05), whereas that of Lactobacillus spp was decreased (Ctrl, 18.05 ± 1.67; HFD, 15.88 ± 8.52; PHPB, 7.07 ± 0.78).

Next, LPS concentration in the serum was measured (Figure 5B) because the effect of PHPB on bacterial species is related to the serum LPS level (LPS acts as an endotoxin). The serum LPS level increased in the HFD group, and was reversed by PHPB treatment (Ctrl, 0.45 ± 0.09; HFD, 0.71 ± 0.15; PHPB, 0.40 ± 0.10).

### 2.6. Effects of PHPB on Macrophage Polarization into M2

To investigate the anti-inflammatory effect of PHPB on macrophage infiltration, macrophages in the colon were stained with F4/80, CD11c, or CD206 antibody (Figure 6A). F4/80 expression was significantly increased in the colon of the HFD group (*p* < 0.05), whereas PHPB treatment reduced F4/80 expression. Furthermore, macrophage polarization into M1 or M2 was characterized with CD11c or CD206 antibody. The HFD group was more positive for CD11c, whereas PHPB treatment resulted in more frequent CD206-positive macrophages. The positive number/400× area of each antibody was quantified with manual counting (Figure 6B). A three-fold increase in F4/80 positive area reflected the increased HFD-induced macrophage infiltration. The infiltrated macrophages were generally differentiated into M1 in the HFD group. However, the PHPB treatment inhibited M1 macrophage differentiation.

### 2.7. Effects of PHPB on Oxidative Stress and ER Stress in Epithelial Cells

The inflammatory environment induced by obesity can cause oxidative and ER stress in goblet cells. To estimate oxidative and ER stress, the gene and protein expression of induced nitric oxide synthase (iNos), cyclooxygenase-2 (Cox2), UPR signaling molecule (sXbp1), ER chaperone (Grp78), and ERAD chaperone (Edem1) in the epithelial cells were analyzed. In Western blot analysis, the normalized intensity values for iNos were Ctrl, 4720.26; HFD, 62,635.34; and PHPB, 10,121.56. For Cox2, they were: Ctrl, 7310.08; HFD, 24,976.47; and PHPB, 8802.13. The PHPB group showed a significantly low iNos and Cox2 synthesis by 2.83-fold and 6.19-fold compared to the HFD group, respectively (Figure 7A). In gene expression analysis, sXbp1 and Grp78 were relatively low, and the expression of Edem1 was relatively high in the PHPB group (Figure 7B).

### 2.8. Effects of PHPB on Goblet Cells

To analyze PHPB effects on goblet cells, H&E, IHC, and Muc2 Ab and PAS/Alcian blue staining were performed (Figure 8). The inhibitory effect of PHPB on oxidative and ER stress was confirmed with histological analysis of the colon, especially goblet cells. When the colon tissues were observed with H&E stain, the HFD group had more severe and widespread colonic damage with frequent crypt loss, mucosa erosion, and an increased number of inflammatory cells in the lamina propria of the intestine compared with that in the Ctrl group. PHPB treatment maintained the tissue characteristics to similar morphologies as the Ctrl group, with few immune cells in the mucosa next to the basal membrane of the epithelial layer (Figure 8A). Furthermore, the HFD group exhibited increased colonic mucosa thickness (612 ± 27.00 μm) compared with that in the Ctrl group (483 ± 46.18 μm). PHPB treatment caused a 13.78% decrease (444 ± 10.39 μm) in the thickness of the mucosa (Figure 8A).

The histologic maintenance by PHPB treatment was confirmed with IHC and PAS/Alcian blue staining. PHPB treatment resulted in the highly frequent appearance of goblet cells (10 ± 3/400× area) within the intestinal crypts as the Ctrl group (13± 2) (Fat group, 4 ± 0.2) (Figure 8B). The results of PAS/Alcian blue staining confirmed the intracellular mucin glycoprotein barrier recovery (Figure 8C).

## 3. Discussion

Obesity and inflammatory bowel disease are co-morbidities. Obesity leads to alteration in lipid metabolism and secretion of inflammatory factors that can induce colon diseases [18]. We investigated several factors associated with obesity-induced colitis, including inflammation. Several researchers have demonstrated that obesity promotes inflammation and contributes to colitis development [19]. Thus, we evaluated the anti-inflammatory effect of PHPB by assessing body/organ weight, food intake, calories, ectopic fat accumulation, lipid metabolism/metabolites, adipogenesis, inflammatory adipokine/cytokine secretion, gut microbiota, serum LPS, macrophage polarization, and oxidative/ER stress. Upon treatment with PHPB, goblet cells maintained histologically and functionally, which ultimately prevents colitis, and this was mediated by the anti-inflammatory effect of PHPB.

We assessed colon and rectum states and observed that diarrhea, stool color, blood or mucus in the stool, and rectal bleeding are symptoms of ulcerative colitis [20]. PHPB treatment improved stool consistency and prevented rectal bleeding. Additionally, the HFD group exhibited shortened colon length, which is one of the typical symptoms of colitis [21]; however, upon PHPB treatment, the colon length was elongated. Furthermore, we assessed the colon weight and observed that the heavier colon weight in the HFD group was reduced by PHPB. The ratio of colon weight to colon length is an indicator of edema in an inflammatory condition [22]. This anti-inflammatory effect of PHPB was also observed in the immunogenic organs, such as the spleen, lung, kidney, and liver. The normalizing of the weights of these organs by PHPB treatment indicates that PHPB can regulate the systemic inflammatory response. Generally, PHPB improved stool consistency, relieved shortened colon, reduced the ratio of colonic weight to length, and improved immunogenic organs.

Furthermore, the effects of PHPB on body weight and food intake were analyzed. Body weight reflects body fat and is used as an obesity indicator when converted into body mass index [2]. HFD caused a significant increase in body weight, which was substantially decreased by PHPB treatment, indicating that PHPB has the ability to induce weight loss. Interestingly, for 10 weeks of observation, this weight loss effect appeared from week 4, which means that a certain period is required for the orally ingested PHPB to show weight loss. In addition to the body weight analysis, we also assessed the gained net weight. The HFD and PHPB groups both received HFD. However, the gained net weight was significantly reduced in the PHPB group, which could be attributed to the difference in the food intake volume. Food intake was similar for the first four weeks in HFD and PHPB groups. However, the PHPB group demonstrated a substantial reduction at week 5, which was maintained for the rest of the period. This reduced food intake indicates that PHPB has an appetite-suppressing effect. Although we did not analyze further, it has been reported that food-derived protein hydrolysates affect the regulation of orexigenic factors in the hypothalamus [23]. The decreased appetite and food intake were evident after determining the calorie equivalent of the food intake. The reduced calorie by PHPB at weeks 5–6 demonstrated a normal diet pattern. These results indicate that PHPB has a weight loss effect through appetite suppression and calorie reduction, which can mediate improvement in obesity-related colitis.

The effect of PHPB on weight control was determined by assessing the abdominal fat. The PHPB group exhibited a significantly reduced visceral, epididymal, and perirenal fat mass. Our analysis focused on abdominal fat (instead of subcutaneous fat) because abdominal fat is considered a more intrinsic component of inflammatory dysregulation than subcutaneous fat [2]. Abdominal fat was quantified by weighing the abdominal adipose tissue, which decreased in the PHPB group. This indicated that PHPB had a hypolipidemic effect on ectopic fats, and this anti-adipogenic effect can lead to clinical improvement mediated by inflammation control. Therefore, the adipogenic pathways were further investigated by assessing systemic lipid metabolism and adipogenesis.

The results of systemic lipid metabolism analysis suggested that PHPB treatment decreased the serum levels of T-chol, TG, and LDL, as well as increased the level of HDL. The systemic lipid metabolism was regulated by the binding of cytokines/adipokines to the receptors on the target cells. Pro-inflammatory molecules increase the serum levels of very-low-density lipoprotein (VLDL), TG, and T-chol, and anti-inflammatory molecules increase the serum level of high-density lipoprotein cholesterol (HDL-C) [24]. The interaction between inflammation and lipid metabolism influences the uptake, transport, synthesis, and degradation through the activities of cytokines/adipokines, indicating that the effect of PHPB in this study was mediated by its anti-adipogenic and anti-inflammatory activities.

The anti-adipogenic effect of PHPB was also demonstrated by the analysis of visceral adipose tissue, which is closely associated with a higher risk of irritable bowel syndrome [25]. To survey the anti-adipogenic mechanism of PHPB, the AMPK signaling pathway was analyzed. The results indicated that PHPB inhibited lipid accumulation, suppressed the expression of peroxisome proliferator-activated receptors (PPARγ) and CCAAT/enhancer binding protein α (C/EBPα), and significantly increased the phosphorylation of AMPKα (p-AMPKα). PPARγ and C/EBPα are transcription factors that can modulate adipocyte maturation26. The activation of the AMPK pathway attenuates PPARγ and C/EBPα expression, consequently inhibiting fat accumulation [26]. This anti-adipogenic effect is connected to the reduction of appetite, body weight, and cellular energy accumulation [26]. Thus, PHPB has anti-adipogenic effects mediated by its ability to reduce adipose tissue volume by inhibiting PPARγ and C/EBPα expression and AMPK phosphorylation. The effects can improve colitis through the inhibition of inflammatory response.

Significant epididymal fat reduction can also be a crucial indicator of anti-inflammatory effect [15]. The epididymis does not have enough space for ectopic fat accumulation [2]; thus, fat accumulation in the epididymis causes inflammatory cell invasion, which promotes inflammatory gene expression and systemic inflammatory responses [15]. Therefore, epididymal fat reduction is also associated with the anti-inflammatory effect of PHPB.

The anti-adipogenic and anti-inflammatory effects of PHPB were investigated by determining cytokine/adipokine levels in the serum, visceral fat, and colon tissue. The results suggested that HFD increased the levels of IL-6, TNF-α, IGF-1, and leptin. The inert storage of excess calories in the adipose tissue is responsible for the increased secretion of pro-inflammatory adipokines [27]. With PHPB treatment, the serum levels of pro-inflammatory molecules were reduced, whereas the serum level of anti-inflammatory adiponectin increased. These results indicate that PHPB has a systemic anti-inflammatory effect. Furthermore, the expression of TNF-α, IL-1β, IL-6, IL-17α, adiponectin, and IL-10 in visceral fat was compared with that in the colon tissue upon PHPB treatment. Both tissues exhibited decreased expression of TNF-α, IL-1β, IL-6, and IL-17α and increased expression of adiponectin, IL-10, defensin, and lipin. These results indicate that PHPB can systemically and locally alter lipid metabolism and regulate inflammatory molecules.

Furthermore, the association between obesity-induced inflammatory cytokines and colitis development was comprehensively analyzed. The closely correlated pro-inflammatory cytokines in lipid metabolism are TNF-a, IL-17a, IL-4, and IL-6 [28,29], and these molecules promote colitis progression [30]. Through the excessive adipocytes at the ectopic adipose tissue, the recruited monocytes activate macrophages, and activated macrophages are the major contributors to inflammatory cytokine production in the gut. Cytokines determine T cell differentiation (Th1, Th2, and T regulatory), and the imbalance of cytokines contributes to the pathogenesis of inflammatory bowel disease [7]. Adiponectin, an anti-inflammatory molecule, prevents acute colitis by restricting B cell immune response [31] and promotes fat oxidation [32]. Furthermore, IL-10 and its receptor prevent spontaneous colitis development [33]. With the analysis of inflammatory molecules, we confirmed that obesity and colitis are closely related through immune cell infiltration and pro-inflammatory cytokines/adipokines. PHPB has an anti-colitic effect mediated by the regulation of inflammatory molecules.

The gut microbiome is another factor associated with obesity-related colitis, as HFD causes dysbiosis, which is related to inflammation and energy consumption. Firmicutes and Bacteroidetes are two dominant gut phyla related to obesity (gram-positive and -negative, respectively) [34]. The balance of the two dominant gut phyla is essential for maintaining homeostasis for immune response and energy balance [34]. The altered gut microbiome exhibited endotoxemia and stool energy harvest change, indicating inflammatory response and weight alterations. Endotoxemia involves a toxic gut environment for Gram-negative bacteria, which is induced by HFD. When Gram-negative bacteria are disrupted, lipopolysaccharide (LPS) is decomposed from the outer membrane of the bacteria cell wall. LPS is a crucial endotoxin, triggering immune cell activation [35]. In colonic fluid, LPS binds to the TLR4 receptors of the intestinal cell membrane and then translocates into the serum. Moreover, LPS binds to immune cells, increases pro-inflammatory cytokines secretion through TLR4 connected to myeloid differentiation primary-response protein 88 (MyD88), and activates the c-Jun N-terminal kinase (JNK) and nuclear factor-kappa B (NFκB) pathways. The representative secreted cytokines are IL-1β, IL-6, and TNF-α [6], and these induce systemic pro-inflammatory responses, including colitis. Our analyses suggested that increased fat intake strongly downregulated the DNA synthesis of Bacteroides and increased the serum LPS levels. As previously reported, the Gram-negative Bacteroides-like intestinal bacteria are highly fragile in plasma with an elevated level of LPS [36]. Our results demonstrated that PHPB treatment enhanced DNA synthesis for Bacteroides-Prevotella and decreased serum LPS levels. These results indicated that PHPB functioned as probiotics by protecting Bacteroides-like intestinal bacteria related to the TLR4-MyD88-JNK/NFκB signaling pathway and reducing LPS in colonic fluid and serum. Therefore, we established that PHPB could inhibit HFD-induced hyperendotoxemia by modulating gut microbiota and reducing serum/colonic fluid LPS levels, eventually improving colitis.

Alteration in gut microbiota also influences energy balance. The possible link between the microbiome and obesity is an obesogenic bacteriologic profile. The obesity-related bacterial species can cause a substantial increase in weight gain compared with the other non-related species [37]. The obesogenic bacteria cause an increase in weight because these bacteria can use undigested foods (or stool) as sources of calories, namely, “stool energy harvest”. In gut microbiota, an elevated level of Firmicutes is associated with obesity, as these bacteria promote fatty acid absorption by inhibiting fatty acid oxidation [38]. Similarly, Lactobacillus is closely associated with body mass through its ability to increase calorie absorption and induce obesity [39]. Our results revealed that Lactobacillus DNA synthesis was significantly reduced by PHPB treatment, which suggests the probiotic-like effect of PHPB against obesogenic gut microbiota. Therefore, we confirm that PHPB can inhibit energy accumulation with stool energy loss by inhibiting obesogenic gut microbiota.

The infiltration, accumulation, and differentiation of macrophages can be directly attributed to obesity-induced colitis. These macrophages usually cause pro-inflammatory M1 polarization in the colon instead of anti-inflammatory M2 polarization [40]. To identify the states of macrophages, we performed IHC with F4/80 and CD11c antibody staining to observe macrophage infiltration rate and M1 polarization, respectively. The PHPB group exhibited reduced F4/80 and CD11c-positive area, which indicates that PHPB can inhibit macrophage infiltration and polarization into M1. Considering that M1 macrophages produce pro-inflammatory cytokines, this result indicates the direct anti-inflammatory effect of PHPB in colitis.

Oxidative and ER stresses were analyzed in the colon. The intestinal epithelial cells directly encounter harmful substances; therefore, these cells could be influenced by various cellular stresses, which leads to the accumulation of misfolded proteins in the ER lumen or ER stress [41]. During protein folding, reactive oxygen species are produced as a by-product. These can disrupt the protein folding mechanism and enhance the production of misfolded proteins, causing further ER stress [42]. Goblet cells are highly sensitive to ER stress [41]. Therefore, to estimate the effect of PHPB on oxidative and ER stresses, iNOS, Cox2, sXbp1, Grp78, and Edem1 were targeted.

For oxidative stress analysis, iNOS and Cox2 protein synthesis were compared, and PHPB treatment indicated inhibitory effects against these enzymes. Considering that the inflamed mucosa in ulcerative colitis produces a high amount of nitric oxide (NO) through inducible COX-2 and iNOS [43], we observed that PHPB treatment has a potential inhibitory effect against oxidative stress, which can be beneficial for the treatment of colitis. The analysis of the expression of ER stress markers sXbp1, Grp78, and Edem1 suggested that PHPB reduced the expression of sXbp1 and Grp78 and increased that of the protein misfolding correction marker (Edem1). The misfolded protein correction reduces ER stress in goblet cells, resulting in the maintenance of the mucosal barrier through normal regulation of MUC2 [44]. Thus, PHPB can regulate oxidative and ER stresses, facilitating the recovery of goblet cells.

The results of the histological analysis of the colon with H&E staining revealed that the HFD group exhibited frequent crypt loss, mucosa erosion, and increased inflammatory cells, which indicated inflammation and ER stress41. IHC with Muc2 Ab staining revealed reduced Muc2 expression in the HFD group, indicating the disruption of goblet cells within the intestinal crypts due to increased pro-inflammatory cytokines and ER stress [41]. Through PAS/Alcian blue staining, PHPB restored mucin glycoprotein recovery within goblet cells. This dense mucus layer can prevent inflammation in the intestine by shielding the epithelium from the external environment. The major macromolecular component of the mucus is the mucin glycoprotein, MUC2, produced by intestinal goblet cells [42]. MUC2 contains highly glycosylated domains and requires extensive post-translational modification within ER and Golgi. The protein complexity of MUC2 and the high secretion property of the goblet cell cause frequent MUC2 protein misfolding, which activates the unfolded protein response (UPR) events and restores protein synthesis in a process known as ER homeostasis. The misfolded protein causes ER stress, inflammatory signaling, apoptosis activation, and suppression of MUC2 transcription [45]. The pale PAS/Alcian blue staining indicates low Muc2 expression, which is caused by UPR-induced suppression of Muc2 transcription. Additionally, the weakened mucus layer increased endotoxin permeability through the mucosal barrier [45], which was demonstrated as an increase in serum LPS (Figure 5B). Collectively, we confirmed that PHPB can inhibit HFD-induced inflammation, oxidative stress, ER stress, goblet cell dysfunction, mucosal barrier disruption, and colitis development.

To further elucidate the mechanisms associated with the anti-colitic effect of PHPB, future studies would involve the pharmacological analysis of exosomes (GM130, CD63, and TSG101), miRNA (miR-155 and miR-34a), free fatty acid (p-ERK, ERK, p-JNK, JNK, p-p38, p38, IκBα, and NF-κB), and lipopolysaccharides (TLR4 and MyD88). The analysis of these pharmacological pathways is vital because each individual has a different response to medication in terms of absorption, distribution, and metabolism. Therefore, a drug that can influence various pathways may be more effective than one that can only influence a single pathway. Obesity is a disease characterized by a chronic inflammatory state in response to various inflammatory factors, such as cancer. Colitis is a disease related to obesity-induced inflammatory response; hence, PHPB, which exhibits anti-inflammatory action through various mechanisms, may be effective against colitis. In addition to analyzing the pharmacological pathways, the dose effects of PHPB must also be analyzed in future studies.

## 4. Materials and Methods

### 4.1. Experimental Design and Diets

The Laboratory Animal Ethics Committee of Yeungnam University approved this experimental protocol (Approval number YUMC-AEC2020-040). The experimental animals (six-week-old male C57BL/6J mice) were obtained from Central Lab. Animal Inc. (Seoul, Republic of Korea) and adapted to a standard diet (Research Diets, New Brunswick, NJ, USA) for one week. The mice were subdivided into three groups of 10 mice each and received the following treatments: (1) Ctrl: normal control mice fed a standard diet (2.93 kcal/g); (2) HFD: mice fed a high-fat diet (rodent diet with 60 Kcal% Fat, 5.24 kcal/g, D12492, Jackson Laboratory, Sacramento, CA, USA); and (3) PHPB: mice fed a high-fat diet and treated with PHPB through the gastric gavage route (16 mg/100 g of body weight/daily) for 10 weeks. The same volume of saline (solvent) was also treated in the HFD group by the same route. The PHPB preparation was prepared as previously described [15]. The body weight, food intake, and general symptoms were monitored during the experiments.

### 4.2. Sampling of Blood, Colon, and Other Target Organs

After an overnight fast, the animals were sacrificed with CO_2_ gas, and the blood and target organs were collected. The blood collected from the heart was centrifuged at 3000 rpm for 15 min to separate the serum, followed by storage at −80 °C until analysis. Subsequently, midline incisions were performed on the mice, the colon was collected, and the length between the cecum and the anus was measured. The fecal content was removed with saline injection, and the weight of the colon was measured after removing the excess water. The visceral/epididymal/perirenal fat tissues, spleen, lung, kidney, and liver were collected and weighed. The colon was divided into three pieces; one was fixed in 10% formalin solution for histopathological analysis, and the others were rapidly frozen with liquid nitrogen and stored at −80 °C for RNA and protein analyses.

### 4.3. Colitis Activity Index

An experienced pathologist modified Cooper’s grading system to assess the disease activity index [17]. The stool consistency was categorized into normal stools (well-formed pellets), loose stools (pasty and semi-formed stools that did not stick to the anus), and diarrhea (liquid stools that stuck to the anus). The occult/gross rectal bleeding was categorized into negative, hemoccult positive, and gross bleeding.

### 4.4. Serum Chemistry

To evaluate the effect of PHPB on systemic inflammation, cytokines/adipokines, tumor necrosis factor-α (TNF-α), interleukin 6 (IL-6), leptin, insulin-like growth factor-1 (IGF-1), adiponectin, and lipopolysaccharide (LPS) concentrations in serum were analyzed using enzyme-linked immunosorbent assay (ELISA) kits (R&D Systems, Minneapolis, MN, USA) according to the manufacturer’s instruction. Furthermore, the hematological effect of PHPB against adipogenesis, total cholesterol (T-chol), triglyceride (TG), high-density lipoprotein (HDL), low-density lipoprotein (LDL), and free fatty acid were analyzed using an automatic biochemical analyzer (Hitachi-720, Hitachi Medical, Tokyo, Japan).

### 4.5. Histopathology and Immunohistochemical (IHC) Analysis

Tissue samples were obtained from the mid colon, fixed in 10% neutral buffered formalin, and embedded in paraffin. The colon sections (5-μm thick) were stained with Periodic acid-Schiff (PAS) and hematoxylin-eosin (H&E). The colitis-induced histological damage was scored with microscopic determination by an expert pathologist according to these criteria: severity of inflammation (none, slight, moderate, and severe); spread of inflammation (none, mucosa, mucosa and submucosa, and transmural); crypt damage (none, basal damaged 1/3, 2/3, only surface epithelium intact, entire crypt, and epithelium lost); and percentage involvement (1–25, 26–50, 51–75, and 75–100%).

For IHC analysis, the colon sections were treated with antigen retrieval citrate buffer, followed by the blocking of endogenous peroxidase and nonspecific binding with a peroxide block and skimmed milk, respectively. The sections were incubated with the primary antibodies at 4 °C overnight, followed by treatment with HRP-labeled anti-rabbit secondary antibody (Sigma-Aldrich, St. Louis, MO, USA) for 1 h at room temperature. The quantitative assessment of the positive area was performed using ImageJ v1.46 (W. Rasband, Open Source). The expression of antigens was evaluated as the percentage of positive pixels estimated over their respective total section area (mm^2^). Antibodies and dilution ratio are presented in Table 1.

### 4.6. Western Blot Analysis

The colon was cut longitudinally (5 mm length), and 10 g of visceral adipose tissue was weighed. The tissue was incubated with 10 mM dithiothreitol in HBSS for 15 min. After removing the supernatant, 0.8 mM EDTA was added, and the mixture was incubated for 30 min with gentle shaking. EDTA was replaced with HBSS, followed by vigorous shaking. The supernatant was centrifuged at 4000× *g* for 5 min, followed by treatment for 20 min on ice with RIPA cell lysate buffer (with complete protease inhibitor cocktail and phosphate stop) and vortexing every 5 min. The lysates were centrifuged at 16,000× *g* for 20 min at 4 °C, and the supernatant was stored at −80 °C. BCA assay was conducted to determine the protein concentration, and the protein was analyzed by using SDS-PAGE analysis. The band density of Western blot was quantified with Image J program (http://wsr.imagej.net/distros/, accessed on 1 July 2022), and the intensity was normalized with β-actin value. The antibodies and their respective dilutions are listed in Table 1.

### 4.7. Gene Expression Analysis

The colon was snap-frozen and homogenized in TRIzol. RNA was extracted using the High Pure RNA Isolation kit (Roche, Basel, Switzerland) according to the manufacturer’s instructions. RNA concentration was measured using a Nanodrop 1000 Spectrophotometer (Thermo Fisher Scientific, Waltham, MA, USA), and complementary DNA was synthesized using 1 μg of RNA with the cDNA synthesis kit (BioRad, Hercules, CA, USA). The expression of genes of interest was analyzed using the One-step SYBR Green PCR kit (Thermo Fisher Scientific) and the Applied Biosystems 7500 Real-Time PCR System (Applied Biosystems, Waltham, MA, USA). PCR involved 40 reaction cycles at 95 °C for 10 s and then 5 s at 95 °C and 30 s at 60 °C. The amplified product was quantified using the comparative cycle threshold (Ct) method, and each sample was normalized with the expression level of GAPDH. Each assay was performed in triplicate. Table 1 lists the primer sequences for the target genes.

### 4.8. Microbial Assessment

For gut microbiota analysis, DNA from cecal contents was extracted using the QIAamp DNA Stool Minikit (QIAGEN, Hilden, Germany) according to the manufacturer’s instructions. After adjusting the DNA concentration to 2.5 ng/μL, amplification was performed with 40 reaction cycles at 95 °C for 5 s, 60 °C for 10 s, and 72 °C for 10 s. Each assay was performed in triplicate. The primers are listed in Table 1.

### 4.9. Statistical Analysis

The values are expressed as mean ± standard deviation. The analysis of variance and post hoc tests were used to assess the data with a statistical significance level of *p* < 0.05. The differences in the assay between the treatment and control groups were determined using the unpaired Student’s *t*-test.

## 5. Conclusions

Collectively, we demonstrated that the oral supplementation of PHPB exhibited inhibitory effects against obesity-related colitis through regulating weight, appetite, fat accumulation, serum lipid metabolites, adipogenesis, inflammatory molecules, gut microbiota, macrophage polarization, oxidative stress, and ER stress. PHPB induced functional and histological recovery in goblet cells. Therefore, we suggest that the protein hydrolysate of Protaetia brevitarsis can improve obesity-related colitis through its anti-inflammatory activities.

## Figures and Tables

**Figure 1 ijms-24-12333-f001:**
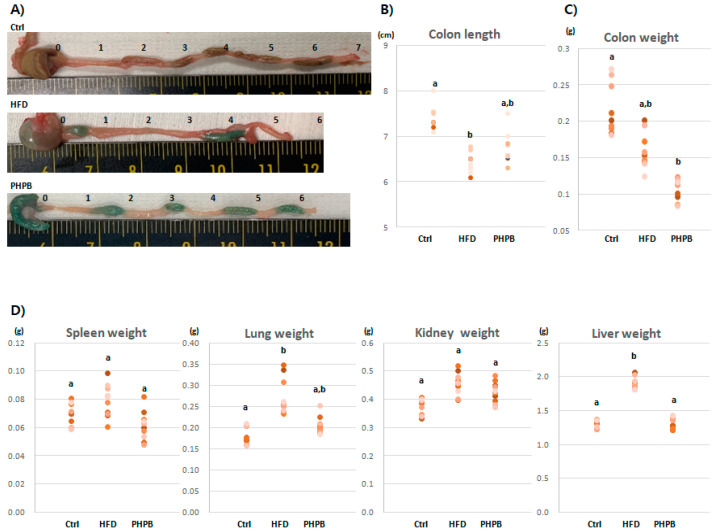
Observation of high-fat-diet-induced colitis and PHPB-mediated alleviation. Gross images of the colon (**A**), comparison of the colon length (**B**) and weight (**C**), and weight changes of individual immune-related organs (**D**). Different letters indicate statistically significant differences (*p* < 0.05).

**Figure 2 ijms-24-12333-f002:**
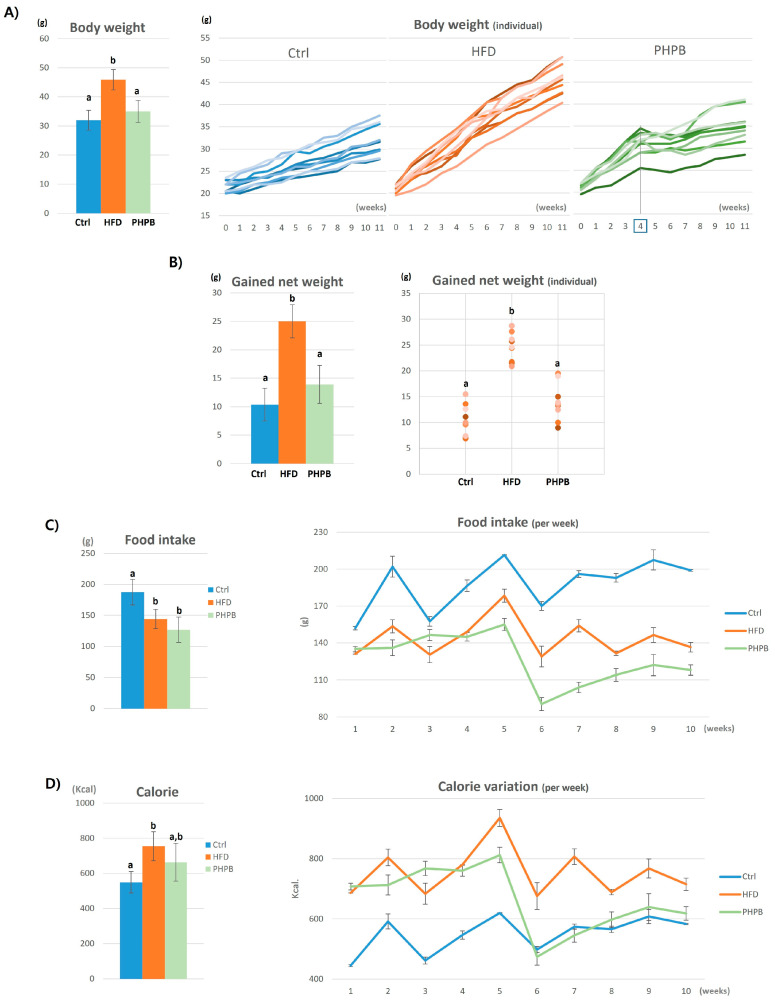
Effects of PHPB on body weight and food intake. Comparison of body weight (**A**), gained net weight (**B**), food intake weight (**C**), and converted calories (**D**). Different letters indicate statistically significant differences (*p* < 0.05).

**Figure 3 ijms-24-12333-f003:**
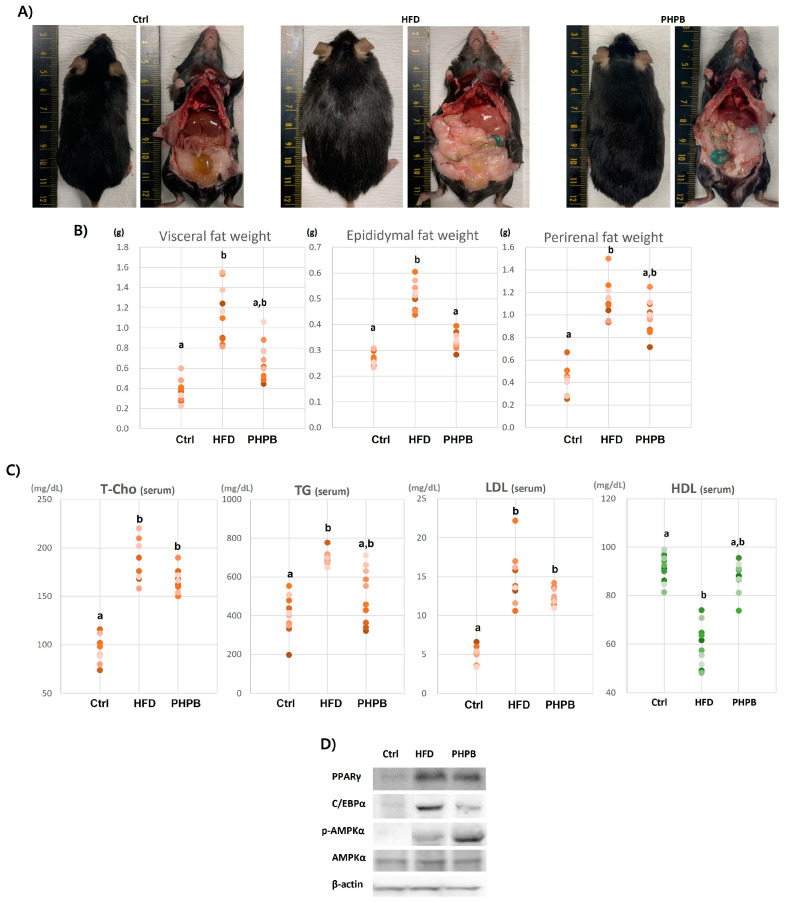
Effects of PHPB on fat accumulation, lipid metabolism, and adipogenesis. The PHPB effects are analyzed through observation of adipose tissue volume in the abdomen (**A**), abdominal fat weight (**B**), serum lipid metabolites level for systemic analysis (**C**), and adipogenesis-related AMPK pathway analysis in visceral adipose tissue (**D**). Different letters indicate statistically significant differences (*p* < 0.05).

**Figure 4 ijms-24-12333-f004:**
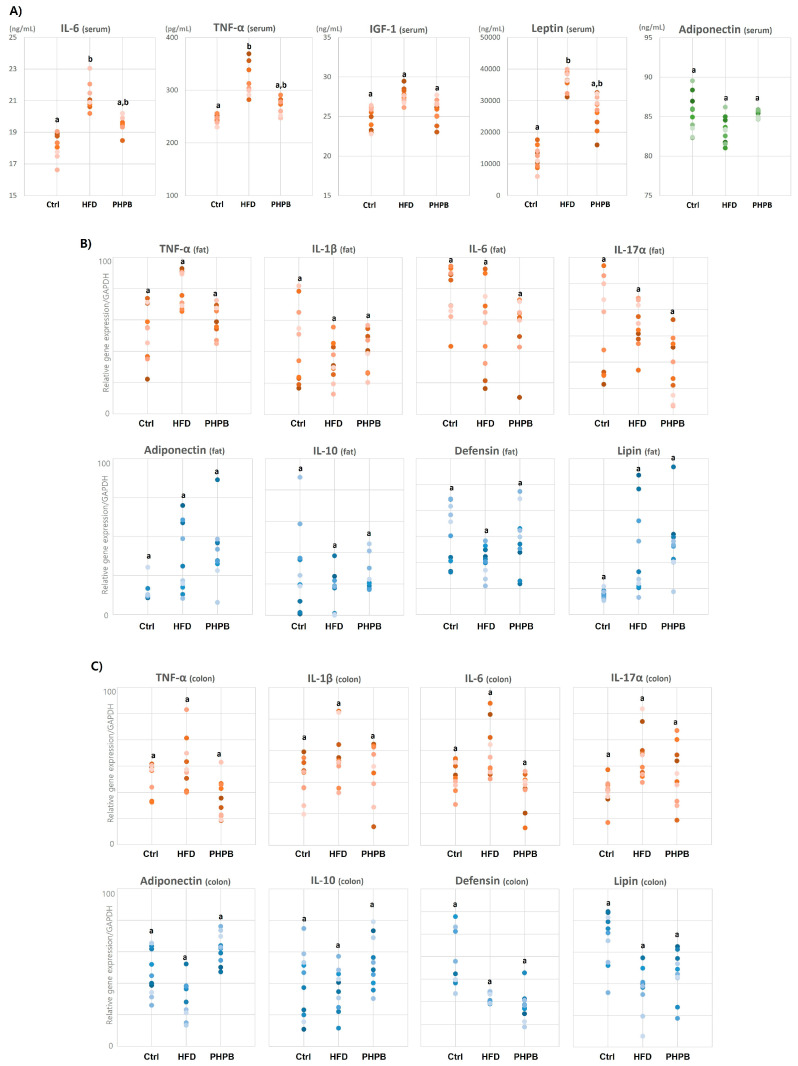
Effects of PHPB on inflammatory molecules. The systemic anti-inflammatory effect of PHPB on cytokines/adipokines in the serum (**A**), visceral adipose tissue (**B**), and colon (**C**). Different letters indicate statistically significant differences (*p* < 0.05).

**Figure 5 ijms-24-12333-f005:**
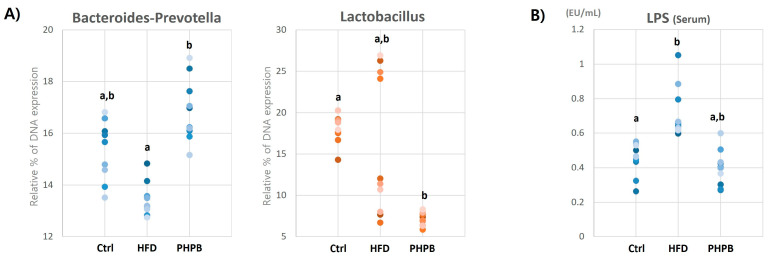
Effects of PHPB on gut microbiota and serum LPS. Relative DNA expression of Bacteroides-Prevotella and Lactobacillus in cecal using real-time PCR analysis (**A**) and serum lipopolysaccharide (LPS) measurement using ELISA (**B**). Different letters indicate statistically significant differences (*p* < 0.05).

**Figure 6 ijms-24-12333-f006:**
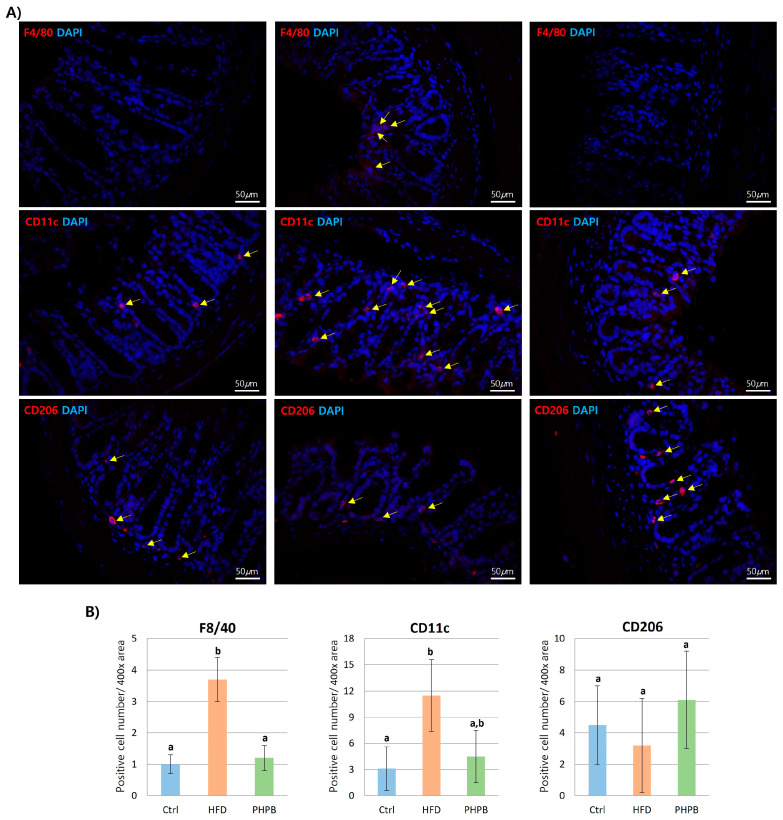
Effects of PHPB on macrophage infiltration and polarization in the colon. Immunohistological analysis with F4/80, CD11c, and CD206 Abs (**A**), and quantification of positive cell number for each Ab (**B**). Different letters indicate statistically significant differences (*p* < 0.05).

**Figure 7 ijms-24-12333-f007:**
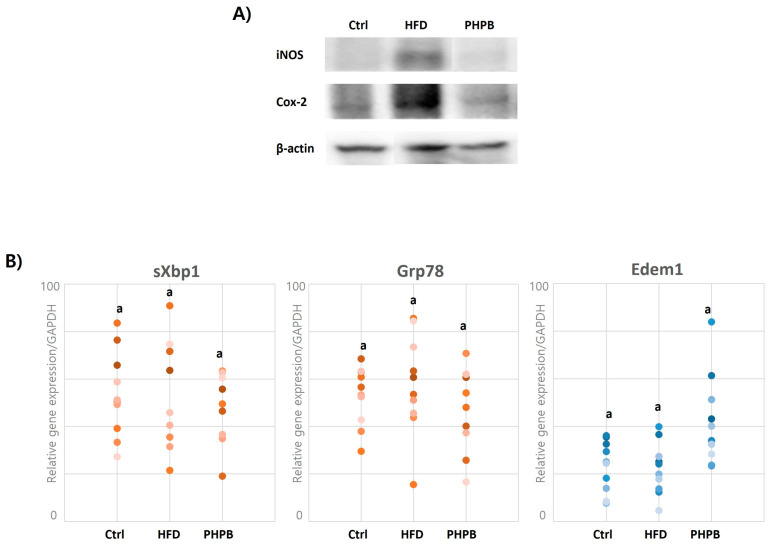
Effects of PHPB on oxidative stress and ER stress. The inhibitory effect of PHPB for oxidative stress markers (iNos and Cox2) (**A**), ER stress positive markers (sXbp1 and Grp78), and misfolded protein correction marker (Edem1) (**B**). The letters indicate statistically significant differences (*p* < 0.05).

**Figure 8 ijms-24-12333-f008:**
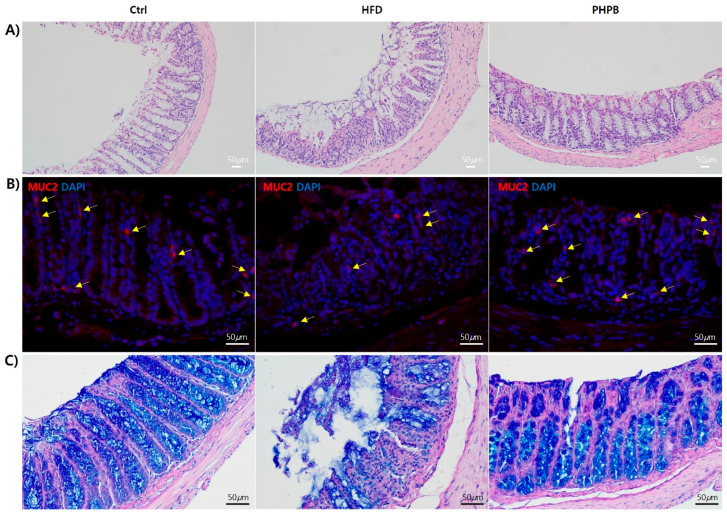
Effects of PHPB on goblet cell recovery. H&E stain (**A**), Immunohistological analysis with Muc2 Ab (**B**), and PAS/Alcian blue staining (**C**) for identification of goblet cell recovery.

**Table 1 ijms-24-12333-t001:** Functional classification, detection method, primer sequences, and antibody information of target molecules.

Functional Classification	Target Molecules	Detection	Primer Seq.	Company, Dilution
Excessive adipogenesis	PPARγ C/EBPα, p-AMPKα AMPKα	WB	-	
Pro-inflammatory cytokine	TNF-α	PCR	F: TTCACTGGAGCCTCGAATGT R: ACCTGACCACTCTCCCTTTG	-
	IL-1β		F: AGGCTCCGAGATGAACAACA R: TCAGCTCATATGGGTCCGAC	
	IL-6		F: AGTTGCCTTCTTGGGACTGA R:TCCACGATTTCCCAGAGAAC	
	IL-17α		F: CAGCGATCATCCCTCAAAGC R: GTCGTTGGCCTCAGTGTTTG	
Anti-inflammatory	Adiponectin	PCR	F: GTTGCAAGCTCTCCTGTTCC R: ATCCAACCTGCACAAGTTCC	-
	IL-10		F: CCAGGG AGATCCTTTGATGA R: AACTGGCCACAGTTTTCAGG	
	Lipin		F: GATGCCGCTAAAGACACTGG R: TGGCTAGGATGCTCACAACA	
	defensin		F: CTGCAAAGGAAGAGAACGCA R: TGGCCTCAGTACTCATGCTC	
Gut microbiota	Bacteroides-Prevotella	PCR	F: GAGAGGAAGGTCCCCCAC R: CGCTACTTGGCTGGTTCAG	-
	Desulfovibrios		F: CCGTAGATATCTGGAGGAACATCAG R: ACATCTAGCATCCATCGTTTACAGC	
	Lactobacillus		F: GAGGCAGCAGTAGGGAATCTTC R: GGCCAGTTACTACCTCTATCCTTCTTC	
	Bifidobacterium		F: CGCGTCTGGTGTGAAAG R: CCCCACATCCAGCATCCA	
	LPS	ELISA	-	R&D Systems
Macrophage polarization	F4/80 CD11c CD206	IHC	-	
Oxidative stress	iNOS Cox2,	WB	-	Abcam, 1:100
ER stress	sXbp1	PCR	F:GAGTCCGCAGCAGGTGC R:CAAAAGGATATCAGACTCAGAATCTGAA	-
	Grp78		F:TGCTGCTAGGCCTGCTCCGA R:CGACCACCGTGCCCACATCC	
	Edem1		F:ATCCTCGGGTGAATCTGAAGACG R:TCATAGAAGGAATCCAGCCCAGC	
Goblet cell dysfunction	Muc2	IHC PAS/Alcian	-	

## Data Availability

All the data used to support the findings of this study are available in the article.

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
