# Peer review of "Protaetia Brevitarsis-Derived Protein Hydrolysate Reduces Obesity-Related Colitis Induced by High-Fat Diet in Mice through Anti-Inflammatory Pathways"

_ijms, 2023, doi:10.3390/ijms241512333_

Round 1

Reviewer 1 Report

Line 35  Abbreviation “HDF” should be introduced in the abstract

Line 40 I suggest to write the organs7cell in which the changes of expression have been detected

Line 49ff The numbering of the references are somehow confusing,  as far as I know the format [1] (with brackets) is recommended.

Line 87 -90 Should be removed

Line 101 Here a “gastroenterologist” is mentioned. I presume it is the “experience pathologist” mentioned in M&M. I suggest to be consistent in the nomenclature.

Line 102. What dextran sulfate sodium treatment is related to. Concentration, duration. Please site the appropriate publication.

Line 103 As far as I understand this paper deals with three different treatments at the same time. Therefore, the PHPB group cannot “improve” the outcome compared to the HFD group. Improve implies, that after HFD treatment the colon length  increases after a followed PHBP treatment. That would have been a good experiment, but that has not been done. Furthermore, giving the Figure 1 B) PHPB is neither different to the Control- nor to the HFD group.

Line 110. You cannot state here that this weight increase is “inflammation-induced”. This is speculative and therefore part of the discussion

Line 113- 115 No weight of any organ has been “restored” and there is no “recovery” here. Please see above.

Line 117-119 Please describe the figures properly. I assume that identical letters within the figure stand for no significant difference??? Why are the dots in Fig b in blue while the others are brown? If you decide the include a Figure weight/mm I suggest to make all directly measured numbers in one color, while the calculated numbers an in another

Line 105ff I cannot quite understand this discussion (well this is the result section). First the weight of the colon in the HDF is not “increased” that that in the PHPH group. It is simply higher (not significantly). As the weight in the control group is even higher the discussion is strange and speculative. Please keep the discussion in the discussion part. If you suggest, that the high weight of the colons within the control group is high due to the length of the organ but I lower per mm of the organ you should calculate that index for all mice.        

Line 145 Please provide a proper legend to the figure. I think that the figures dealing with weight can be reduced without loosing any information, the figures regarding the food intake can be remove at this compares incomparable parameters (the food is not identical).

Line 151, 155 , 157, 162 again “reduced”, “ reduction” “decreased” I will stop now to make comments on that. I hope you got the point

Line 167-168 That is again discussion

Line 189-190 Again Discussion. I will not comment on that topic again

Line 192-193 Again Legend to the figure must improve. I will neither comment on that any more

 Line 228 How has the mentioned quantification been made

Line 267ff I do not understand this sentence

Line 275 In the investigation PHBS was not given after the colitis has developed due to the high fat diet. Therefore, PHBS does not improve colitis. It at most prevent colitis.

Line 270 after the term “several researchers” I expect more than one citation. Or a recommendation for a recent review.

Line 296 The explanation that oral administered anti- inflammatory PHPB is not working for three weeks needs more explanation. Orally administered drugs (ASS) works within 30 minutes. I might well be that PHBS modifies the microbiome and that subsequently the effects are observable, but this needs more and detailed discussion.    

Line 301 This is an over interpretation of the data. All groups show the effect. So, I think there was some changed in the maintenance/handling of the animals.

Line 469 There I no evidence, that PHPB exhibits anti-inflammatory actions though various mechanisms. I well could be that the mentioned reducing in appetite causes the reported findings. This could be investigated by adding a control group with restricted calories after high fat diet. As there are some reports on that (i.E doi: 10.1139/apnm-2020-0220; doi: 10.2337/db11-1621) a detailed and profound discussion on that point seems to be sufficient.    

The discussion over all I mainly a repetition of the result section. I am missing discussion on the anti-inflammatory effect of extracts of Protaetia brevitarsis seulensis which have been described already in the literature and some profound an literature based discussion on the pathway(s) of PGPB

Line 483 If the other two groups are not treated by a gastric gavage the experiments is not clear focused on the PHPB treatment. The manipulation with the gastric gavage with the solvent of the PHPB should be performed on a separate group. If that is not possible in short time a statement should be given that the treatment with the gastric gauge does not influence the outcome (supported by finding in the literature)

Line 498 and line 516  Was the “experiences pathologist” blind to the experimental group of the rats

Line 535 and 537  Are The centrifugal forces (40 x g, 16 x g ) correct?

Some minor adaptations are required

Author Response

Reviewer #1

Line 35  Abbreviation “HDF” should be introduced in the abstract

The Abbreviation for “HDF” in the abstract was revised into high-fat diet (HFD) (revised version, line 43).

Line 40 I suggest to write the organs&cell in which the changes of expression have been detected

According to the reviewer's suggest, colon organ was filled in the text ) (revised version, line 48).

Line 49 The numbering of the references are somehow confusing,  as far as I know the format [1] (with brackets) is recommended.

The numbering of the references are corrected to fit the format.

Line 87 -90 Should be removed

We deleted the sentence you pointed out, thank you for the careful review.

Line 101 Here a “gastroenterologist” is mentioned. I presume it is the “experience pathologist” mentioned in M&M. I suggest to be consistent in the nomenclature.

For the consistency of the nomenclature, “gastroenterologist” is changed into "pathologist” (revised version, line 107).

Line 102. What dextran sulfate sodium treatment is related to. Concentration, duration. Please site the appropriate publication.

This sentence is misplaced, so it has been removed.

Line 103 As far as I understand this paper deals with three different treatments at the same time. Therefore, the PHPB group cannot “improve” the outcome compared to the HFD group. Improve implies, that after HFD treatment the colon length  increases after a followed PHBP treatment. That would have been a good experiment, but that has not been done. Furthermore, giving the Figure 1 B) PHPB is neither different to the Control- nor to the HFD group.

We agree your point, we changed the expression from "improve" into "less shorten" (revised version, line 108).

Line 110. You cannot state here that this weight increase is “inflammation-induced”. This is speculative and therefore part of the discussion

We agree your point. The expression " An inflammation-induced increase in colon weight was also observed in other immune-related organs." is changed into " Like colon weight, other immune-related organs also showed weight change patterns.." (revised version, line 117).

Line 113- 115 No weight of any organ has been “restored” and there is no “recovery” here. Please see above.

We agree your point, and the words "restored" and "recovery" were replaced into "decrease". That sentence changed into "With PHPB treatment, the weights of spleen, lung and kidney were slightly decreased (0.06 ± 0.01, 0.20 ± 0.02, 0.43 ± 0.04, and 1.3 ± 0.09 g, respectively) compared to the HFD group, and liver weight was significantly decreased."

Line 117-119 Please describe the figures properly. I assume that identical letters within the figure stand for no significant difference???

Line 192-193 Again Legend to the figure must improve. I will neither comment on that any more

Different letters indicate statistically significant differences (P<0.05). This phrase puts into all figure legends.

Why are the dots in Fig b in blue while the others are brown?

The intention of the different colors was to distinguish length and weight. However, if the reader confuses, a single color seems to be correct. Thus, figure 1 B's dot color was repainted into brown.

If you decide the include a Figure weight/mm I suggest to make all directly measured numbers in one color, while the calculated numbers an in another

To express each object, the dot color expressed in gradation. However, If the reviewer does not agree with this, we will mark it in one color.

Line 105 I cannot quite understand this discussion (well this is the result section). First the weight of the colon in the HDF is not “increased” that that in the PHPH group. It is simply higher (not significantly). As the weight in the control group is even higher the discussion is strange and speculative.

We agree your point, and the sentence was changed into " The HFD-induced colitis was also evidenced by the colon weight (Fig. 1C). The proportion of length and weight of the colon is related to inflammatory response or edema. The HFD group exhibited increased weight (0.16 ± 0.02 g) than that in the PHPB group (0.11 ± 0.01 g)." (revised version, line 110).

Please keep the discussion in the discussion part.

Line 167-168 That is again discussion

Line 189-190 Again Discussion. I will not comment on that topic again

All discussions in the results section have been removed.

If you suggest, that the high weight of the colons within the control group is high due to the length of the organ but I lower per mm of the organ you should calculate that index for all mice.       

Of course, the longer the length, the heavier it is, but the HFD group showed a heavier colon weight than the short colon length compared to the PHPB group. Thus, we suggest the edema in the HFD group.

Line 145 Please provide a proper legend to the figure.

The figure legend was modified into " Fig. 2. Effects of PHPB on body weight and food intake. Comparison of body weight (A), gained net weight (B), food intake weight (C), and converted calories (D)."

I think that the figures dealing with weight can be reduced without loosing any information, the figures regarding the food intake can be remove at this compares incomparable parameters (the food is not identical).

We agree with the opinion that there is no need for a graph of food intake because the food is not identical. In fact, mice do not like high-fat diets. The reason why we included this graph is to 1) visually show the decrease in high-fat feed intake, suggesting the effect of reducing appetite by PHPB as the cause, and then, 2) to show the decrease in calories according to the decrease in intake. Although the experimental groups ate the same high-fat diet, the reason for the decrease in intake in the PHPB group is appetite, we pointed that in the discussion (original version, line 136). However, if the reviewer does not agree above the reason, we will remove the figure 2 C).

Line 151, 155 , 157, 162 again “reduced”, “ reduction” “decreased” I will stop now to make comments on that. I hope you got the point

All the expressions that you mentioned changed into "low", " volume", " inhibited", "maintained" or "prevent" depending on the context of the sentence.

Line 228 How has the mentioned quantification been made

The counting method was described in the text "The positive number/400× area of each antibody was quantified with manual counting" (revised version, line 227).

Line 267ff I do not understand this sentence

It is an error in the correction process, we removed this sentence.

Line 275 In the investigation PHBS was not given after the colitis has developed due to the high fat diet. Therefore, PHBS does not improve colitis. It at most prevent colitis.

We agree your point, that word changed into "prevent" (revised version, line 285).

Line 270 after the term “several researchers” I expect more than one citation. Or a recommendation for a recent review.

As you mentioned, the reference was changed into the review paper "19.           Kreuter R, Wankell M, Ahlenstiel G, et al. The role of obesity in inflammatory bowel disease. Molecular Basis of Disease 2019; 1865:63-72. 2018/10/20. DOI: 10.1016/ j.bbadis.2018.10.020."

Line 296 The explanation that oral administered anti- inflammatory PHPB is not working for three weeks needs more explanation. Orally administered drugs (ASS) works within 30 minutes. I might well be that PHBS modifies the microbiome and that subsequently the effects are observable, but this needs more and detailed discussion.   

We agree your point. The time it takes for oral medications to produce an anti-inflammatory effect is very short, less than 30 minutes. Our intention in this paragraph is how long it will take for the weight loss effect to appear, which takes about 4 weeks. To avoid confusion for the reader, the sentence has been modified as follows: Interestingly, for 10 weeks of observation, this weight loss effect appeared from week 4, which means that a certain period is required for the orally ingested PHPB to show weight loss (revised version, line 304).

Line 301 This is an over interpretation of the data. All groups show the effect. So, I think there was some changed in the maintenance/handling of the animals.

Although we did not analyzed further, it has been reported that food-derived protein hydrolysates can affect the regulation of orexigenic factors in the hypothalamus. Thus, the sentence was modified by adding references (revised version, line 312).

Line 469 There I no evidence, that PHPB exhibits anti-inflammatory actions though various mechanisms. I well could be that the mentioned reducing in appetite causes the reported findings. This could be investigated by adding a control group with restricted calories after high fat diet. As there are some reports on that (i.E doi: 10.1139/apnm-2020-0220; doi: 10.2337/db11-1621) a detailed and profound discussion on that point seems to be sufficient.    

This study mainly analyzes the inflammatory response. We will explore the mechanisms involved in calorie restricted for future studies.

The discussion over all I mainly a repetition of the result section. I am missing discussion on the anti-inflammatory effect of extracts of Protaetia brevitarsis seulensis which have been described already in the literature and some profound an literature based discussion on the pathway(s) of PHPB

This is the first paper to prove the anti-inflammatory effect of PHPB on colitis. It is also the first time that these effects have been demonstrated in 11 pathways. Since the anti-inflammatory mechanisms of protein hydrolysates are very diverse, the results of the 11 mechanisms presented in our paper are expected to make a significant contribution to demonstrating the anti-inflammatory effect of PHPB on colitis.

Line 483 If the other two groups are not treated by a gastric gavage the experiments is not clear focused on the PHPB treatment. The manipulation with the gastric gavage with the solvent of the PHPB should be performed on a separate group. If that is not possible in short time a statement should be given that the treatment with the gastric gauge does not influence the outcome (supported by finding in the literature)

The same volume of saline (solvent) was also treated in the HFD group by the same route. We added this sentence into phase (revised version, line 497).

Line 498 and line 516  Was the “experiences pathologist” blind to the experimental group of the rats

Of course, the blind test was conducted.

Line 535 and 537  Are The centrifugal forces (40 x g, 16 x g ) correct?

there were mistakes, there are corrected into "4,000" and "16,000" (revised version, line 549, 551).

We appreciate all of your kind mentions.

Reviewer 2 Report

Authors used Protaetia brevitarsis-derived protein hydrolysate to reduce the obesity-related colitis induced by high-fat diet in mice through anti-inflammatory pathways. The paper has the following shortcomings:

1. Please unify the format of "MUC2, Muc2, Muc-2" and "Cox-2, COX-2, Cox2" in the full text and tables.

2. L87: Please check the format.

3. L360: correct "IL-628".

4. L386: correct " TNF-α6".

5. L487: correct "CO2".

6. All Western Blot results in this paper do not visually analyze protein bands, which makes it impossible for readers to intuitively judge the experimental results.

7. In Fig4, except serum IL-6, TNF-α, Leptin, all inflammatory factors in serum, visceral adipose tissue and colon did not show significant significance. How to prove that “PHPB has local and systemic anti-inflammatory effects”.

8. None of the three figures in Figure 7B shows significance, how in proving that “PHPB has an antioxidant effect and can reduce ER stress”.

      9. Several linguistic and typo errors are apparent and must be revised carefully.

Author Response

Reviewer 2

  1. Please unify the format of "MUC2, Muc2, Muc-2" and "Cox-2, COX-2, Cox2" in the full text and tables.

Depending on the context, for human proteins, all capital letters are used, and for mouse proteins, only the first letter is capitalized. The hyphens were corrected. Thank you for your careful comments.

  1. L87: Please check the format.

As also pointed out by reviewer #1, it was corrected (revised version, line 97).

  1. L360: correct "IL-628".

Due to a typo according to the reference format, it is corrected (revised version, line 373).

  1. L386: correct " TNF-α6".

It is also a typo, and revised (revised version, line 400).

  1. L487: correct "CO2".

This was changed to subscript (revised version, line 487).

  1. All Western Blot results in this paper do not visually analyze protein bands, which makes it impossible for readers to intuitively judge the experimental results.

To intuitively judge the Fig. 3D and Fig.7A results, the band density of Western blot was quantified with Image J program, and the intensity was normalized with β-actin value. The detailed values were inserted into the sentences (revised version, line 170, 242).

  1. In Fig4, except serum IL-6, TNF-α, Leptin, all inflammatory factors in serum, visceral adipose tissue and colon did not show significant significance. How to prove that “PHPB has local and systemic anti-inflammatory effects”.
  2. None of the three figures in Figure 7B shows significance, how in proving that “PHPB has an antioxidant effect and can reduce ER stress”.

The PHPB used in this study is not a purified drug phase, is more crude. Therefore, although it is not a significant result, we think that such a trend was shown. Future studies will be conducted with more purified substances to obtain more significant results.

  1. Several linguistic and typo errors are apparent and must be revised carefully.

All errors were corrected with careful revision.

We appreciate all of your kind mentions.

Round 2

Reviewer 1 Report

Thank you for answering the raised issues.

The English Language is understandable.There are only some minor flaws.

Reviewer 2 Report

It can be accepted in the present form.